# Intracranial Compliance Assessed by Intracranial Pressure Pulse Waveform [note 1]

**DOI:** 10.3390/brainsci11080971

**Published:** 2021-07-23

**Authors:** Sérgio Brasil, Davi Jorge Fontoura Solla, Ricardo de Carvalho Nogueira, Manoel Jacobsen Teixeira, Luiz Marcelo Sá Malbouisson, Wellingson Silva Paiva

**Affiliations:** 1Department of Neurology, School of Medicine, University of São Paulo, São Paulo 05508-070, Brazil; davisolla@hotmail.com (D.J.F.S.); rcnogueira28@gmail.com (R.d.C.N.); manoeljacobsen@gmail.com (M.J.T.); wellingson.paiva@hc.fm.usp.br (W.S.P.); 2Department of Intensive Care, School of Medicine, University of São Paulo, São Paulo 05508-070, Brazil; luiz.malbouisson@hc.fm.usp.br

**Keywords:** intracranial compliance, intracranial pressure, decompressive craniectomy, acute brain damage

## Abstract

Background: Morphological alterations in intracranial pressure (ICP) pulse waveform (ICPW) secondary to intracranial hypertension (ICP >20 mmHg) and a reduction in intracranial compliance (ICC) are well known indicators of neurological severity. The exclusive exploration of modifications in ICPW after either the loss of skull integrity or surgical procedures for intracranial hypertension resolution is not a common approach studied. The present study aimed to assess the morphological alterations in ICPW among neurocritical care patients with skull defects and decompressive craniectomy (DC) by comparing the variations in ICPW features according to elevations in mean ICP values. Methods: Patients requiring ICP monitoring because of acute brain injury were included. A continuous record of 10 min-length for the beat-by-beat analysis of ICPW was performed, with ICP elevation produced by means of ultrasound-guided manual internal jugular vein compression at the end of the record. ICPW features (peak amplitude ratio (P2/P1), time interval to pulse peak (TTP) and pulse amplitude) were counterweighed between baseline and compression periods. Results were distributed for three groups: intact skull (exclusive burr hole for ICP monitoring), craniotomy/large fractures (group 2) or DC (group 3). Results: 57 patients were analyzed. A total of 21 (36%) presented no skull defects, 21 (36%) belonged to group 2, whereas 15 (26%) had DC. ICP was not significantly different between groups: ±15.11 for intact, 15.33 for group 2 and ±20.81 mmHg for group 3, with ICP-induced elevation also similar between groups (*p* = 0.56). Significant elevation was observed for the P2/P1 ratio for groups 1 and 2, whereas a reduction was observed in group 3 (elevation of ±0.09 for groups 1 and 2, but a reduction of 0.03 for group 3, *p* = 0.01), and no significant results were obtained for TTP and pulse amplitudes. Conclusion: In the present study, intracranial pressure pulse waveform analysis indicated that intracranial compliance was significantly more impaired among decompressive craniectomy patients, although ICPW indicated DC to be protective for further influences of ICP elevations over the brain. The analysis of ICPW seems to be an alternative to real-time ICC assessment.

## 1. Introduction

The skull content—the cerebrospinal fluid (CSF), the brain, and the blood volumes—is a major component and determinant of intracranial pressure (ICP). The capacity to accommodate the different intracranial compartments is named intracranial compliance (ICC). ICC is a property of dynamic volumes inside a cavity whose expansion is very limited, indicating the hemostasis amongst them [1]. In addition, ICC may reflect the compensatory changes of vessels (mainly the great intracranial venous sinuses) and the CSF spaces (cisterns, ventricles and subarachnoid), according to ICP elevations [2,3,4].

Recently, the International Multidisciplinary Consensus Conference on Multimodality Monitoring in Neurocritical Care made a list of recommendations that included ICP monitorization [5]. The consensus strongly recommends ICP monitoring to guide medical and surgical interventions, and to detect life-threatening imminent herniation. Nevertheless, the ICP threshold value for intervention is still uncertain. The continuous assessment and monitoring of ICP, including waveform quality, is also strongly recommended [6]. In addition, the committee indicated that further research into the relationship between ICP and clinical outcomes will benefit from automated, high-resolution monitoring and alternate forms of analysis [5].

Likewise, in recent years the knowledge on the ICP pulse waveform (ICPW) has advanced, as well as its clinical application. ICPW is an early marker of ICC impairment [3,7,8,9] and is mainly represented by three distinct peaks: P1 (percussion wave), P2 (tidal wave) and P3 (dicrotic wave) [10]. Under physiologic conditions, P1 produced by arterial contraction is the highest peak observed, with P2 reflecting both vascular and ventricular repercussion of pressure pulse spread. As cerebrovascular resistance is normally lower [11] in comparison with other systems and organs, the tidal wave assumes an amplitude lower than P1. When the buffering mechanisms described above are exhausted and intracranial hypertension (ICH) is present, there is [12] deformation of ICPW, with P2 assuming an amplitude higher than P1, and the ICPW becomes progressively pyramidal [13], with the enlargement of the time interval between P1 and P2 [7] (Figure 1).

All the findings described above have been investigated in animal models or clinical observational studies; however, ICPW changes after ICH treatment surgery have been poorly assessed. Surgery for mass lesion removal or decompressive craniectomy (DC) changes brain architecture and dynamics [14], which makes ICP thresholds also change. In this environment, multimodal monitoring becomes essential to guide therapeutic interventions [6,15,16]. Although neurosurgical procedures are effective for ICP control, morbidity remains high, which might be explained by the persistence of low ICC (despite ICP under acceptable standards). The primary objective of the present study was to evaluate ICPW changes after surgery procedures and correlate them with ICC.

## 2. Methods

This is a single center, prospective and observational case series trial in the neurological intensive care unit of Hospital das Clínicas, São Paulo University, Brazil. This clinical trial (CT) study protocol was approved by the local Ethics Committee, on 23 May 2017 (REB register 66721217.0.0000.0068), and registered under number NCT03144219 (available from 15 July 2017 at clinicaltrials.gov). All methods were performed in accordance with the relevant guidelines and regulations, and informed consent was obtained from all legally authorized representatives (LAR)/next of kin instead of the patients because of illness severity.

### 2.1. Study Design

All patients included in the study suffered an acute brain injury with the need for ventilatory support and invasive ICP monitoring in accordance with guidelines adopted by our institution. Data collection consisted of a 10-min session of simultaneous recording of spontaneous fluctuations of invasive arterial blood pressure, ICP, heart rate and oxygen. At minute 7, an ultrasound-guided manual internal jugular vein compression (IJVC) was performed for 60 s (Figure 2).

### 2.2. Participants

Inclusion criteria consisted of any neurocritical patients who underwent ICP invasive monitoring up to the fifth day of catheter insertion. We excluded those presenting with fixed mydriatic or middle-sized pupils for more than 2 h after ventilatory and hemodynamic stabilization.

### 2.3. Clinical and Intracranial Variables

Demography, clinical, imaging presentation and severity scores were recorded. The clinical variables collected were age in years (continuous variable), diagnostics, admission Glasgow score, simplified acute physiologic score (SAPS3), Marshall tomographic score in the case of traumatic brain injury (TBI), modified Fisher tomographic score in the case of subarachnoid hemorrhage (SAH), arterial blood pressure, axillar temperature, heart and respiratory rates, oxygen saturation and sedatives administrated. ICP was monitored with the Neurovent monitoring system (Raumedic^®^, Munchberg, Germany), which consists of a pressure probe for ventricular use. This system can be attached to any monitor using a small zero-point specific simulator for the patient monitor type.

### 2.4. Data Acquisition and Analysis

The automated analytics system verified all data collected, i.e., ICP pulse wave morphology parameters such as the P2/P1 ratio (P2 amplitude divided for P1 amplitude), the time-to-peak (TTP—time interval from the beginning of each pulse until P2) interval and pulses amplitudes (mean amplitudes of each pulse) [17]. For this study, all calculations were performed using the mean pulse of the ICP, excluding possible artifacts. The mean pulse was obtained by calculating the amplitudes of the P1 and P2 peaks and subtracting to the base value of the ICP pulse. The automated system was Phyton based, with Numpy and Scipy libraries [18]. This system calculated the time interval where P2 should be depicted on the waveform and TTP according to the cardiac cycle (Figure 3).

### 2.5. Sample Size

The sample size was not calculated a priori, but the achieved sample size had 80% power to detect interactions between the three skull groups (intact, craniotomy/fracture and craniectomy) and the two moments (before and during jugular vein compression) with an effect size (η^2^) of at least 0.06 (moderate), assuming alpha error probability 0.05 and correlation among repeated measures 0.5.

### 2.6. Statistical Analysis

For descriptive purposes, categorical variables were presented through relative and absolute frequencies and compared using the chi-squared or Fisher exact test, as appropriate. Continuous variable distributions were deemed normal as assessed by skewness, kurtosis and graphical methods. There were no missing data for the intracranial monitoring parameters.

Repeated measures ANOVA analyses were employed to compare the intracranial monitoring parameters’ behavior between the groups along the experiment. The effect size was standardized by the eta squared (η^2^). As a sensitivity analysis, a multivariable linear regression was modeled to verify the effect of the skull defect in the intracranial parameter variation (during compression—baseline) after adjustment for age and the baseline parameter.

All tests were 2-sided and final *p* values under 0.05 were considered statistically significant. All analyses were conducted with SPSS software (IBM Corp. SPSS Statistics para Windows, version 24.0. Armonk, NY, USA).

## 3. Results

### Sample Features

A total of 98 eligible, consecutive patients admitted between August 2017 and May 2020 were included. A total of 41 patients were excluded from this analysis—7 were not adults, 25 recordings disclosed not proper quality for analysis and for 10 patients data were lost. A total of 38,456 intracranial pressure pulses were analyzed, derived from the 10-min sessions of our final sample of 57 patients (Figure 4).

Table 1 depicts the sample characteristics according to skull defect groups. Age, sex, hemoglobin and general clinical severity (SAPS3) were similar between groups. Regarding the pathology, stroke was more frequent among those with craniectomy (adjusted residual 3.4, *p* = 0.028). DC was left side for eight (53%) patients, right side for six (40%) and bilateral in one (6%) case.

Table 2 presents the intracranial monitoring parameters according to skull defect and adjusted for age. All groups had an ICP increase during IJVC, but no interaction was disclosed between group and period (baseline/compression) (*p* = 0.565) (Figure 5). The P2/P1 ratio also increased during IJVC for the intact and craniotomy/fracture groups, but did not change for the craniectomy group (*p* value for interaction 0.010 and partial η^2^ 0.161, a large effect size). Time-to-peak and amplitude did not change significantly during IJVC nor presented interaction between group and period (baseline/compression). These results are the same after adjustment for hemoglobin and SAPS3.

Since baseline ICP and the P2/P1 ratio tended to be higher in the craniectomy group, sensitivity analyses were conducted to verify the independent association between this skull defect and the P2/P1 ratio behavior during IJVC. Figure 6 presents a stratified analysis of the P2/P1 ratio behavior during IJVC according to baseline P2/P1 status (normal or altered). Those with an altered baseline P2/P1 ratio and intact skull (*n* = 6 (28%)) or craniotomy/fracture (*n* = 10 (47%)) presented an increase in the P2/P1 ratio during IJVC (mean difference 0.05, 95% IC 0.01–0.09, *p* = 0.033), but not those with craniectomy (*n* = 9 (60%); mean difference −0.07, 95% IC −0.15–0.02, *p* = 0.103; *p* value for interaction 0.026). Thus, the results presented in Table 2 and Figure 6 cannot be attributed to a ceiling effect. Similarly, those with a normal baseline P2/P1 ratio and intact skull (*n* = 15) or craniotomy/fracture (*n* = 11) presented an increase in the P2/P1 ratio during IJVC (mean difference 0.10, 95% IC 0.05–0.16, *p* = 0.001), but not those with craniectomy (*n* = 6; mean difference 0.03, 95% IC −0.02–0.08, *p* = 0.134), although the interaction did not reach statistical significance (*p* = 0.425).

Aiming to perform another sensitivity analysis, a multivariable linear regression was modeled to verify the effect of the skull defect in the P2/P1 ratio variation (during compression—baseline) after adjustment for age and the baseline parameter (Table 3). Compared to the intact or craniotomy/fracture groups, craniectomy was independently associated with less P2/P1 ratio variation (−0.09, 95% IC −0.15–−0.02, *p* = 0.009), regardless of the baseline status.

No adverse events were reported during this study’s intervention and monitoring.

## 4. Discussion

The main novelty of the present study was to explore ICPW features in a population of neurosurgical patients when skull integrity has been lost. The main finding was the observation of an opposite behavior regarding the P2/P1 ratio among subjects with acute closed skull brain damage and even after craniotomy, in comparison with the craniectomized patients. Although not statistically significant, DC patients disclosed higher baseline mean ICP levels and P2/P1 ratio, with a significant decrease in the latter when ICP elevation was induced. Thus, the study demonstrated that the P2/P1 ratio may bring additional information for neuromonitoring [3,19] and the effect of DC that not just ameliorated ICP but also increased the intracranial compliance, protecting for further elevations in ICP. This agrees with previous studies using different methodologies [20,21,22].

Patients submitted to DC have a more severe neurological condition, and that is the reason why the ICP was higher and the P2/P1 ratio remained higher after DC in comparison to other groups. Interestingly, despite the higher ICP values, our study demonstrated greater ICC in this group as observed with the P2/P1 ratio after an induced rise in ICP [23,24]. This phenomenon has never been demonstrated before and should be further explored as an indicator of successful DC.

DC is an effective procedure to alleviate extremely high ICP, although the evidence of efficacy amongst different neurological pathologies is variable [25,26]. For most diseases, an ICP threshold of 19 mmHg has been associated with good outcome; nevertheless, lower ICP values may improve survival [27]. Notwithstanding, a decrease in mortality has not been proven, especially because of difficulties of elaborating studies with this subject [15].

The improvement in cerebral perfusion after DC is associated with favorable outcomes [20,21] and with the prevention of metabolic crisis [28,29]. This association was evaluated by Jin et al. in 60 DC patients, suggesting thresholds for predicting good prognosis in DC according to ICP values (ICP <19 mmHg in the first 24 h) and transcranial Doppler (TCD) derived parameters (mean blood flow velocity >56.33 cm/s, end-diastolic blood flow velocity >40.28 cm/s, and resistance index <0.57) [30]. Moreover, Lubillo et al. assessed 42 DC patients and observed that “changes in brain oximetry before and after DC, measured with probes in non-injured brain have independent prognostic value for the 6-month outcome in TBI patients” [31].

Previous studies have focused either on the extraction of indexes or ICP areas under curve calculations to obtain measures of ICC [10]. Unfortunately, the applicability of ICP waveform-derived information to date is mostly restricted because of the need for specialized hardware and software, making these observations and findings less present in daily practice. Timofeev et al. [32], using dedicated software (ICM+, University of Cambridge, Cambridge, UK), studied ICP waveform amplitude in correlation with mean ICP values, the RAP index, and the correlation between arterial blood pressure and ICP, the PRx. Likewise, Asgari et al. [33], with another dedicated software program (MOCAIP, University of California at Los Angeles, Los Angeles, CA, USA), performed automated ICP peak analysis during cerebrovascular changes. However, the opportunity of a non-invasive bedside observation of ICC has been recently developed (B4C, São Carlos, Brazil) and was validated in children with hydrocephalus [34] and severe COVID-19 cases [35,36]. This system provides the ICPW in real time, with automated P2/P1 ratio calculation. Further investigations may validate the use of this technology as a screening tool for patients with progressive ICC deterioration who should undergo decompressive craniectomy.

Regarding critical care, new studies are highlighting the importance of the neurological surveillance of inadvertent events that may worsen patients’ prognosis [37,38], whilst multimodality in neurocritical care for the acquisition of brain metabolism, electrical activity, oxygenation, cerebral perfusion and ICP probably cover all patients’ needs, with real hindrances relying on the limitations and reliability of each technique itself. Altogether, with reference to ICP, this advice has shown the importance of considering not only the mean values of ICP in mmHg, but also the characteristics of the ICPW, which combines markers of both cerebral hemodynamics and cerebrospinal pressure–volume compensation that encodes information about the biophysical characteristics of the intracranial space [39,40,41].

### Limitations

Although IJV compression is a maneuver able to be performed at the bedside, and thus reproducible in clinical sets, this is not an ICP controlled measure, with considerable variation being observed among patients. The results observed in the present study were with reference mostly to slight variations in ICP because stimulating higher elevations would be considered unethical. Thus, it is not possible to predict ICPW behavior with reference to substantial elevations in ICP. ICP values before DC were not reported, although as patients were included in their first days after ICP catheter implantation, significant SAPS3 heterogeneity between groups would preclude comparison, which did not happen. This was a cross-sectional observation; thus, clinical outcome measures were not suitable to calculate in accordance with our results.

## 5. Conclusions

Intracranial pressure pulse waveform is a reliable marker of intracranial compliance and may play a role besides intracranial pressure mean values for the neurocritical patient. After decompressive craniectomy, further elevations in ICP did not lead to additional deterioration in intracranial compliance.

## Figures and Tables

**Figure 1 brainsci-11-00971-f001:**
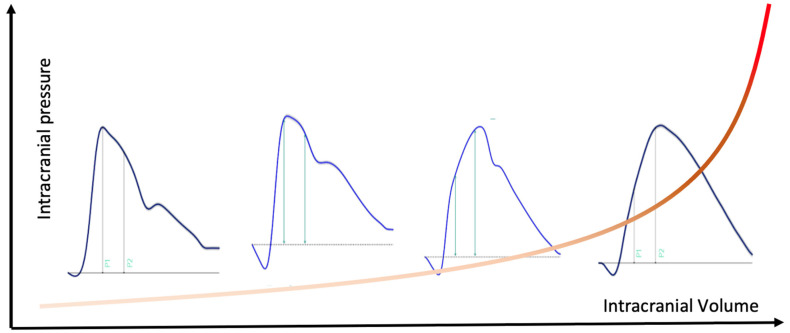
Langfitt’s pressure/volume wave superimposed with ICP waveforms disclosing that as intracranial compliance reduces, a more pyramidal shape is assumed (progressive P2/P1 ratio).

**Figure 2 brainsci-11-00971-f002:**
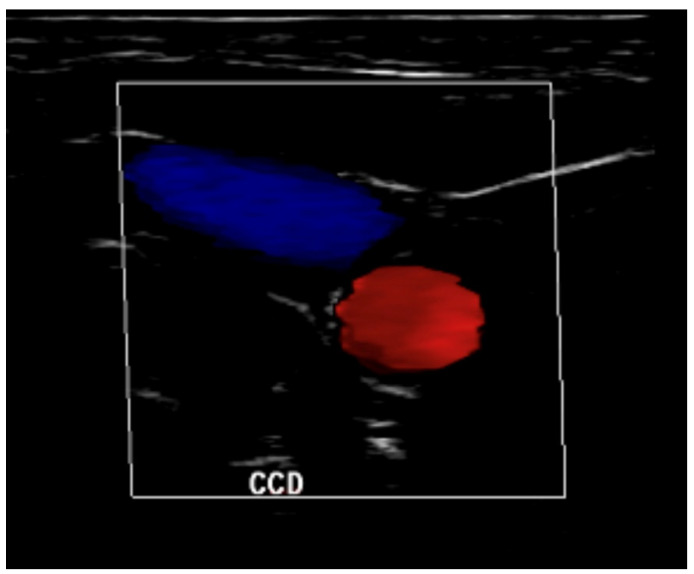
A manual internal jugular vein (blue) compression was performed for 60 s to elevate ICP with the aid of ultrasound to avoid compressing common carotid artery (red).

**Figure 3 brainsci-11-00971-f003:**
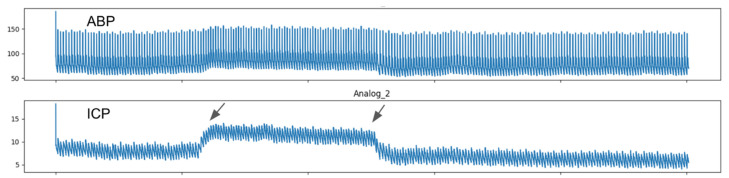
Mean ICP, P2/P1, TTP and pulse amplitude were calculated and correlated between baseline and 60 s IJV compression (plateau between arrows) intervals.

**Figure 4 brainsci-11-00971-f004:**
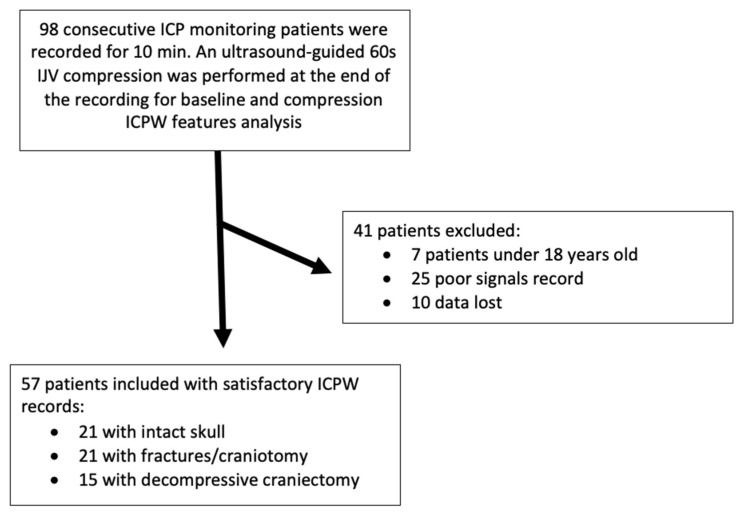
Flow chart with sample and procedure description. ICP: intracranial pressure, ICPW: intracranial pressure pulse waveform, IJV: internal jugular vein.

**Figure 5 brainsci-11-00971-f005:**
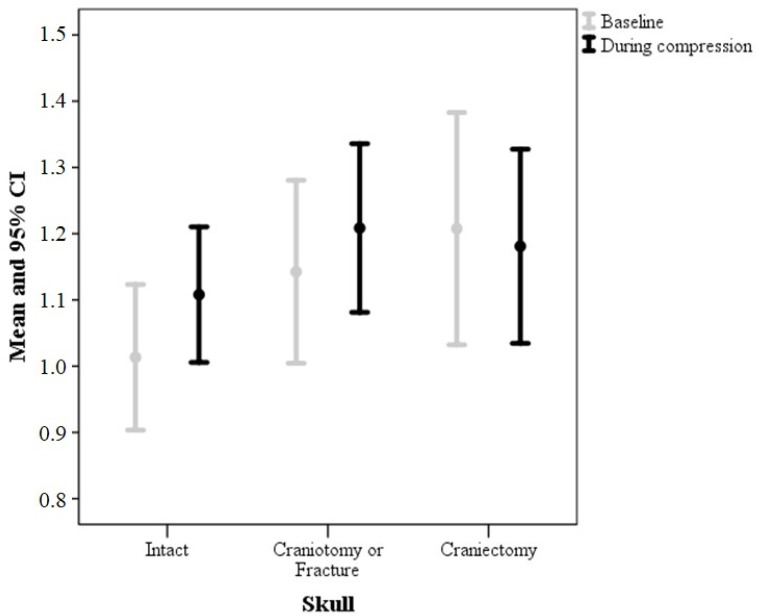
P2/P1 ratio according to skull defect.

**Figure 6 brainsci-11-00971-f006:**
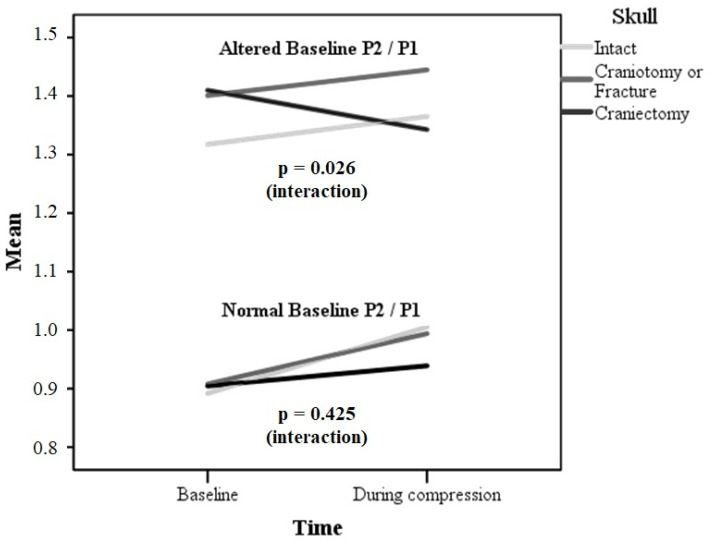
P2/P1 according to skull defect and baseline P2/P1 status.

**Table 1 brainsci-11-00971-t001:** Sample characteristics according to skull defect (*n* = 57).

Variable	Skull			*p*-Value
	Intact (21)	Craniotomy or Fracture (21)	Craniectomy (15)	
**Age**	38.3 ± 15.2	36.1 ± 11.1	36.5 ± 13.8	0.551
**Male sex**	15 (71.4)	10 (47.6)	12 (80.0)	0.098
**Pathology**				0.028
Traumatic brain injury	17 (81.0)	14 (66.7)	9 (60.0)	
Subarachnoid hemorrhage	4 (19.0)	5 (23.8)	1 (6.7)	
Stroke	0 (0.0)	1 (4.8)	5 (33.3)	
Tumor	0 (0.0)	1 (4.8)	0 (0.0)	
**Hemoglobin**	10.3 ± 1.6	10.0 ± 1.5	10.3 ± 1.8	0.854
<10 mg/dL	8 (38.1)	9 (42.9)	6 (40.0)	0.951
<9 mg/dL	5 (23.8)	4 (19.0)	4 (26.7)	0.857
<8 mg/dL	2 (9.5)	1 (4.8)	0 (0.0)	0.447
**SAPS3**	53.5 ± 12.1	62.2 ± 12.6	63.2 ± 14.3	0.082
**Admission GCS**	8.4 ± 4.3	7 ± 3.3	6.7 ± 3.2	0.093
**BMI**	22.5 ± 4.2	22.9 ± 3.7	21.7 ± 2.5	0.143
**30 days mortality**	5 (23)	6 (28)	5 (33)	0.075

Categorical variables are presented as *n* (%). Continuous variables are presented as mean ± standard deviation. GCS: Glasgow coma score, BMI: body mass index.

**Table 2 brainsci-11-00971-t002:** Intracranial monitoring parameters according to skull defect (*n* = 57).

Parameter	Skull	Baseline	Jugular Vein Compression	Difference (95% CI)	*p*-Value	Partial η^2^
Intracranial pressure (mmHg)	Intact	15.11 ± 8.10	19.45 ± 7.65	4.54 (3.22–5.87)	0.565	0.021
	Craniotomy or Fracture	15.33 ± 6.53	19.62 ± 7.44	3.90 (2.90–4.91)		
	Craniectomy	20.81 ± 10.22	23.93 ± 9.46	2.44 (1.64–3.24)		
P2/P1 ratio	Intact	1.01 ± 0.24	1.11 ± 0.22	0.09 (0.04–0.15)	0.010	0.161
	Craniotomy or Fracture	1.14 ± 0.30	1.21 ± 0.28	0.07 (0.02–0.11)		
	Craniectomy	1.21 ± 0.32	1.18 ± 0.26	−0.03 (−0.8–0.03)		
Time-to-peak (ms)	Intact	0.20 ± 0.08	0.21 ± 0.08	0.01 (−0.01–0.03)	0.693	0.014
	Craniotomy or Fracture	0.23 ± 0.09	0.25 ± 0.08	0.02 (−0.01–0.05)		
	Craniectomy	0.22 ± 0.10	0.23 ± 0.10	0.01 (−0.01–0.03)		
Amplitude (mV)	Intact	10.87 ± 7.80	11.35 ± 7.79	0.48 (−0.06–1.03)	0.739	0.011
	Craniotomy or Fracture	5.58 ± 3.32	6.06 ± 3.92	0.48 (0.07–0.90)		
	Craniectomy	4.31 ± 3.09	4.56 ± 3.32	0.25 (−0.17–0.67)		

Data presented as mean ± standard deviation. The *p* values refer to the interaction between skull defect and time. Adjusted for age. CI: Confidence interval.

**Table 3 brainsci-11-00971-t003:** Multivariable linear regression for P2/P1 variation (after compression—baseline).

Variable	Coefficient (95% CI)	SE	Standardized Coefficient	*t* Value	*p* Value
Craniectomy (compared to intact or craniotomy/fracture)	−0.09 (−0.15–−0.02)	0.03	−0.33	−2.72	0.009
Altered baseline P2/P1 ratio	−0.07 (−0.13–−0.01)	0.03	−0.30	−2.49	0.016
Age	−0.001 (−0.002–0.001)	0.001	−0.20	−1.74	0.088

CI: Confidence interval; SE: Standard error.

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
