# Peer review of "Intracranial Compliance Assessed by Intracranial Pressure Pulse Waveform†"

_brainsci, 2021, doi:10.3390/brainsci11080971_

Round 1

Reviewer 1 Report

Dear Authors, I congratulate you for this interesting study. Neurocritically ill patients represent a complex model of brain pathophysiology that needs still to be largely studied.

Intracranial pressure monitoring is a long debate history among these patients, and benefit or harm are still under investigation. However, I strongly believe that ICP monitoring gives important informations, and in the context of multimodal neuromonitoring, it takes its place!

I have some queries and suggestion about your study that I would like you to explain me.

The first time when you use an abbreviation, you should include also the extended form for the word.

Please, whenever you present data in a table, always put the unit of measure (for example, in table 2, I cannot find the unit of measure for ICP, and so on...)

Please, include a study flow-chart.

In the table 1, in the cohort of patients with intact skull, you reported 35.2±22.8 as mean age. Would it imply that also patients < 18 years were enrolled? If it is so, I have some doubts about the uniformity of the sample included in the study (you probably know that children have different ICC than adults, and also different values of threshold for ICP treatment!).

Please specify what type of software analysis for ICP wave and derived parameters (P2/P1 ratio for instance)  that you employed for the study in the methods section.

Sample size calculation: I don't understand if a sample size was calculated or not...

Table 2: what do you mean "adjusted for age"? 

What were ICP values in the group of patients before decompressive craniectomy?

How were performed DC? Emicraniectomy? Bilateral? Else?

I would be interested in receiving a revised version of your study.

Best regards

Author Response

The first time when you use an abbreviation, you should include also the extended form for the word.

This has been checked throughout the text.

Please, whenever you present data in a table, always put the unit of measure (for example, in table 2, I cannot find the unit of measure for ICP, and so on...)

Units were added, as mmHg for ICP, ms for TTP and mV for pulse amplitude, please refer to page 9

Please, include a study flow-chart.

Included on page 8

In the table 1, in the cohort of patients with intact skull, you reported 35.2±22.8 as mean age. Would it imply that also patients < 18 years were enrolled? If it is so, I have some doubts about the uniformity of the sample included in the study (you probably know that children have different ICC than adults, and also different values of threshold for ICP treatment!).

Age results were mistakenly taken from the entire sample (98 patients). The analysis in this study was particular for adults, although children have been enrolled in this project to the analysis of cerebral hemodynamics that will be explored in another study. We reanalyzed age and corrected table 1, please check.

Please specify what type of software analysis for ICP wave and derived parameters (P2/P1 ratio for instance)  that you employed for the study in the methods section.

Analysis software and reference were added, page 6.

Sample size calculation: I don't understand if a sample size was calculated or not...

This was better explained, page 7

Table 2: what do you mean "adjusted for age"? 

By "adjusted for age", we meant that age was introduced as a covariate in the repeated measures ANOVA model (a procedure called ANCOVA by some authors). In other words, the results are independent of the patient's age.

What were ICP values in the group of patients before decompressive craniectomy?

We didn’t report this and include it as a limitation. Patients were included in their first days of ICP implantation, as such, disease severity was adjusted for admission SAPS3, being no significant admission disease severity difference between groups.

How were performed DC? Emicraniectomy? Bilateral? Else?

This information was included at the end of page 8.

Reviewer 2 Report

TITLE: Intracranial compliance assessed by intracranial pressure pulse waveform.

Dear Authors, I found this study innovative and interesting. I would add some comments and ask some questions before moving forward.

The aim of this study was to assess the morphological alterations in intracranial pressure pulse waveform (ICPPW) among neurocritical care patients with and without decompressive craniectomy (DC), by comparing the variations of ICPPW features according to elevations in mean ICP values. Elevation in ICP was produced by means of ultrasound-guided manual internal jugular veins compression. During the analysis of data, patients were divided into three groups: intact skull (exclusive burr hole for ICP monitoring), craniotomy/large fractures (group 2) or DC (group 3).

Abstract: overall clear but needs some improvements:

  • The study design should be included.

The results showed that. A total of 57 patients were analyzed. 21 (36%) presented no skull defects, whereas 15 (26%) hadDC.

  • If you mentioned 3 groups for a total of 57 patients, you should describe in the results how many patients were allocated in the 3 groups (not only in 2 groups). You stated that 21 had no skull defects, and 15 skull defects. What about the other 21?

ICP was not significantly different between groups: ±13.59 for intact and ±17.66 mmHg for DC, with ICP induced elevation also similar between groups (p= 0.56).

  • Yu are describing what happened between two groups, what about AMONG the three groups? One group is missing here.

Significant elevation was observed for P2/P1 ratio for groups 1 and 2, whereas reduction was observed in group 3 (elevation of ±0.09 for groups 1 and 2, whereas reduction of 0.03 for group 3, p=0.01).

  • Finally, you described the group 3: ok.

Introduction: ok

Methods:

  • Table 1 should be improved with other informations on the patients (death/ comorbidities, weight, BMI, etc etc). Why did you focus on hemoglobin?
  • I would suggest adding “early effect” in the methods and the title and the type of study design, when clearly defined.
  • You assessed a maneuver to rise ICP, but you did not argue after how many times you evaluated the waveforms. I suppose you did it immediately after (if not maybe the ethical value should be argument).
  • Uncontrolled Clinical trial? Is the design appropriately described? The primary objective of a clinical trial is to demonstrate the efficacy of a new treatment. In clinical trials he subjects in the control group receive another active treatment/ maneuver/ management or placebo, if ethically justified. Thus, randomized controlled trials are the gold standard of clinical research, whereas uncontrolled trials are considered as sources of very weak clinical evidence. One regards uncontrolled trials that are born as uncontrolled by necessity, i.e., inclusion of a control is technically unfeasible or has no rational basis (i.e., set up of new methods (e.g., measurements of abdominal or gastric or bladder pressure, localization of prostate during radiotherapy, or monitoring the success of an intervention, regardless of its efficacy on the underlying disease (e.g., survival of a transplanted organ or a decompressive craniectomy).

I would argue that the method you used, although innovative, is not the first time is described. The ICPPW changes after ICH treatment surgery has never been assessed. Thus, is not the method the novelty, but the population you evaluated.

Don’t you think that an observational design would be more appropriated? For sure, it would be simplest to justify some lacking methodological features for uncontrolled trials. It seems that you performed a pre-post experimental study to test the effect of jugular compression.

I kindly ask the authors to justify their choice and to better argue this point in their manuscript.

  • Statistical analysis: appropriate
  • Discussion: appropriate, I would suggest modifying “To our knowledge, this is the first study on the relations of ICP peaks observing the P2/P1 ratio, time-to-peak and amplitudes var- iations among neurocritical patients with a slight provoked ICP elevation”. Many new studies are coming out with prospective observational/ pre-post design on this topic but with different patients observed. (“Effects of passive leg raising test, fluid challenge and norepinephrine on cerebral autoregulation in COVID-19 critically ill patients” Frontiers Neurol 2021. Early effects of ventilatory rescue therapies on systemic and cerebral oxygenation in mechanically ventilated COVID-19 patients with acute respiratory distress syndrome: a prospective observational study. Crit Care 2021) that should be argument in the discussion.
  • Lacking some limitations.

Author Response

Abstract: overall clear but needs some improvements:

The study design should be included.

Abstract study design was improved, please check

The results showed that. A total of 57 patients were analyzed. 21 (36%) presented no skull defects, whereas 15 (26%) hadDC. If you mentioned 3 groups for a total of 57 patients, you should describe in the results how many patients were allocated in the 3 groups (not only in 2 groups). You stated that 21 had no skull defects, and 15 skull defects. What about the other 21?

Group 2 sample was included

ICP was not significantly different between groups: ±13.59 for intact and ±17.66 mmHg for DC, with ICP induced elevation also similar between groups (p= 0.56). You are describing what happened between two groups, what about AMONG the three groups? One group is missing here.

The results were corrected in accordance with table 2, group 2 was also included here

Significant elevation was observed for P2/P1 ratio for groups 1 and 2, whereas reduction was observed in group 3 (elevation of ±0.09 for groups 1 and 2, whereas reduction of 0.03 for group 3, p=0.01). Finally, you described the group 3: ok.

Introduction: ok

Methods:

Table 1 should be improved with other informations on the patients (death/ comorbidities, weight, BMI, etc etc). Why did you focus on hemoglobin?

Further variables were added, hemoglobin could reflect higher blood loss for brain hemorrhages and DC in comparison with intact skull patients, this was not significant instead.  

I would suggest adding “early effect” in the methods and the title and the type of study design, when clearly defined. You assessed a maneuver to rise ICP, but you did not argue after how many times you evaluated the waveforms. I suppose you did it immediately after (if not maybe the ethical value should be argument).

The record sessions were short to avoid inclusion of considerable variations in main parameters that could include bias, as arterial pressure, heart rate and respiratory rate per example. The analysis beat-by-beat of ICP pulses allowed the automated system to obtain means of each ICPW parameter, and then compare the spontaneous baseline period with the ICP raised period, so the analysis of 38456 ICP pulses obtained from our patients was quite significative  

Uncontrolled Clinical trial? Is the design appropriately described? The primary objective of a clinical trial is to demonstrate the efficacy of a new treatment. In clinical trials the subjects in the control group receive another active treatment/ maneuver/ management or placebo, if ethically justified. Thus, randomized controlled trials are the gold standard of clinical research, whereas uncontrolled trials are considered as sources of very weak clinical evidence. One regards uncontrolled trials that are born as uncontrolled by necessity, i.e., inclusion of a control is technically unfeasible or has no rational basis (i.e., set up of new methods (e.g., measurements of abdominal or gastric or bladder pressure, localization of prostate during radiotherapy, or monitoring the success of an intervention, regardless of its efficacy on the underlying disease (e.g., survival of a transplanted organ or a decompressive craniectomy).

Don’t you think that an observational design would be more appropriated? For sure, it would be simplest to justify some lacking methodological features for uncontrolled trials. It seems that you performed a pre-post experimental study to test the effect of jugular compression. I kindly ask the authors to justify their choice and to better argue this point in their manuscript.

Thank you for the valuable explanation and suggestions, the design was changed for case series, as patients were included prospectively according to spontaneous hospital admissions. The analysis of ICPW allowed to divide three different groups according to skull situation, although no control group was included. Regarding compressing the IJVs, the results demonstrated that this maneuver is effective to elevate ICP.

Statistical analysis: appropriate

Discussion: appropriate,

I would argue that the method you used, although innovative, is not the first time is described. The ICPW changes after ICH treatment surgery has never been assessed. Thus, is not the method the novelty, but the population you evaluated.

This has been better highlighted on the first discussion’s paragraph

I would suggest modifying “To our knowledge, this is the first study on the relations of ICP peaks observing the P2/P1 ratio, time-to-peak and amplitudes variations among neurocritical patients with a slight provoked ICP elevation”. Many new studies are coming out with prospective observational/ pre-post design on this topic but with different patients observed. (“Effects of passive leg raising test, fluid challenge and norepinephrine on cerebral autoregulation in COVID-19 critically ill patients” Frontiers Neurol 2021. Early effects of ventilatory rescue therapies on systemic and cerebral oxygenation in mechanically ventilated COVID-19 patients with acute respiratory distress syndrome: a prospective observational study. Crit Care 2021) that should be argument in the discussion.

These references have been added in the context of new insights on the use of neurological multimodality for non-neurological patients

Lacking some limitations.

I have discussed limitations a little more, please check.

Round 2

Reviewer 1 Report

Thank you, you addressed all my queries.